# Correlation between Autofluorescence Intensity and Histopathological Features in Non-Melanoma Skin Cancer: An Ex Vivo Study

**DOI:** 10.3390/cancers13163974

**Published:** 2021-08-06

**Authors:** Ilaria Giovannacci, Marco Meleti, Federico Garbarino, Anna Maria Cesinaro, Ema Mataca, Giuseppe Pedrazzi, Camilla Reggiani, Alessia Paganelli, Arianna Truzzi, Federica Elia, Luca Giacomelli, Cristina Magnoni

**Affiliations:** 1Department of Dermatology, Surgical, Medical and Dental Department of Morphological Sciences Related to Transplant, Oncology and Regenerative Medicine, University of Modena and Reggio Emilia, 41124 Modena, Italy; ilaria.giovannacci@gmail.com (I.G.); garbarinofederico2@gmail.com (F.G.); camilla.reggiani@libero.it (C.R.); alessia.paganelli@gmail.com (A.P.); 2Centro Universitario di Odontoiatria, Department of Medicine and Surgery, University of Parma, 43121 Parma, Italy; marco.meleti@unipr.it; 3PhD Program in Clinical and Experimental Medicine, University of Modena and Reggio Emilia, 41124 Modena, Italy; 4Department of Anatomic Pathology, University of Modena and Reggio Emilia, 41124 Modena, Italy; cesinaro.annamaria@policlinico.mo.it (A.M.C.); mataca.ema@gmail.com (E.M.); 5Department of Medicine and Surgery and Robust Statistics Academy, University of Parma, 43121 Parma, Italy; pedrazzi@unipr.it; 6School of Dentistry, University of Modena and Reggio Emilia, 41124 Modena, Italy; ariannatruzzi@gmail.com (A.T.); fedeelia95@gmail.com (F.E.); 7Polistudium SRL, 20124 Milan, Italy; luca.giacomelli@polistudium.it

**Keywords:** autofluorescence, NMSC, optical biopsy, dermatologic surgery

## Abstract

**Simple Summary:**

Non-melanoma skin cancers (NMSC) are the most prevalent neoplasms worldwide, especially in the Caucasian population. Radical surgical excision is considered the therapeutic gold standard, while incomplete tumor removal is invariably associated with recurrence and the need for reintervention. Autofluorescence (AF) spectroscopy has recently been investigated for early diagnosis of NMSC and assessment of tumor margins. Understanding changes in AF intensity in association with peculiar histological features could improve the diagnostic accuracy of skin fluorescence spectroscopy. The main goal of our work was to investigate the correlation between the intensity of cutaneous AF and the histopathological features of NMSC. The intensity of fluorescence emission in tissues obtained from NMSC samples was approximately 4 times lower than that in healthy conditions. In fact, mean AF intensity for BCC group was 4.5 and 4.4 for SCCs, with further variability being recorded according to histopathologic subtypes. Histopathological features such as hyperkeratosis, neoangiogenesis, fibrosis and elastosis are statistically associated with a decrease in AF intensity. Our data suggest that such tissue alterations could be responsible for the difference in AF emission between neoplastic and healthy tissue. These results support the potential application of AF as a useful non-invasive diagnostic tool for NMSCs.

**Abstract:**

Non-melanoma skin cancer (NMSC) is the most common malignant tumor affecting fair-skinned people. Increasing incidence rates of NMSC have been reported worldwide, which is an important challenge in terms of public health management. Surgical excision with pre-operatively identified margins is one of the most common and effective treatment strategies. Incomplete tumor removal is associated with a very high risk of recurrence and re-excision. Biological tissues can absorb and re-emit specific light wave-lengths, detectable through spectrophotometric devices. Such a phenomenon is known as autofluorescence (AF). AF spectroscopy has been widely explored for non-invasive, early detection of NMSC as well as for evaluation of surgical margins before excision. Fluorescence-aided diagnosis is based on differences in spectral characteristics between healthy and neoplastic skin. Understanding the biological basis of such differences and correlating AF intensity to histological features could improve the diagnostic accuracy of skin fluorescence spectroscopy. The primary objective of the present pre-clinical ex vivo study is to investigate the correlation between the intensity of cutaneous AF and the histopathological features of NMSC. Ninety-eight lesions suggestive for NMSCs were radically excised from 75 patients (46 M; 29 F; mean age: 79 years). After removal, 115 specific reference points on lesions (“cases”; 59 on BBC, 53 on SCC and 3 on other lesions) and on peri-lesional healthy skin (controls; 115 healthy skin) were identified and marked through suture stitches. Such reference points were irradiated at 400–430 nm wavelength, and resulting emission AF spectra were acquired through spectrophotometry. For each case, AFIR (autofluorescence intensity ratio) was measured as the ratio between the number of photons emitted at a wavelength ranging between 450 and 700 nm (peak: 500 nm) in the healthy skin and that was captured in the pathological tissue. At the histological level, hyperkeratosis, neoangiogenesis, cellular atypia, epithelial thickening, fibrosis and elastosis were quantified by light microscopy and were assessed through a previously validated grading system. Statistical correlation between histologic variables and AFIR was calculated through linear regression. Spectrometric evaluation was performed on 230 (115 cases + 115 controls) reference points. The mean AFIR for BCC group was 4.5, while the mean AFIR for SCC group was 4.4 and the fluorescence peaks at 500 nm were approximately 4 times lower (hypo-fluorescent) in BCCs and in SCCs than in healthy skin. Histological variables significantly associated with alteration of AFIR were fibrosis and elastosis (*p* < 0.05), neoangiogenesis, hyperkeratosis and epithelial thickening. Cellular atypia was not significantly associated with alteration of AFIR. The intensity of fluorescence emission in neoplastic tissues was approximately 4 times lower than that in healthy tissues. Histopathological features such as hyperkeratosis, neoangiogenesis, fibrosis and elastosis are statistically associated with the decrease in AFIR. We hypothesize that such tissue alterations are among the possible biophysical and biochemical bases of difference in emission AF between neoplastic and healthy tissue. The results of the present evaluation highlighted the possible usefulness of autofluorescence as diagnostic, non-invasive and real-time tool for NMSCs.

## 1. Introduction

Non-melanoma skin cancer (NMSC) represents the most common cancer in fair-skin people. Basal cell carcinoma (BCC) accounts for approximately 75% of cases, the remaining mainly being squamous cell carcinomas (SCC) [1,2]. While BCC rarely metastasizes, metastases from high-risk SCCs are not uncommon and may be fatal [3,4,5,6,7].

Management of NMSC is based on early diagnosis and treatment, as well as monitoring of recurrences [8,9,10,11].

Several techniques, based on optical properties of skin, and including dermoscopy, reflectance confocal microscopy (RCM), multiphoton microscopy, fluorescence evaluation, diffuse reflectance and Raman spectroscopy, have been variously used as tools for non-invasive diagnosis of NMSCs [12,13,14]. Among these, dermoscopy and RCM are reliable techniques for melanoma diagnosis even if their efficacy for preoperative delineation of SCC surgical margins is questionable [15,16,17,18].

Autofluorescence (AF) is a type of luminescence depending on the optical properties of specific molecules (fluorophores) able to absorb and re-emit light of specific wavelengths.

Alteration of fluorescence (hyper and/or hypofluorescence) has been documented in neoplastic tissues, presumptively resulting from changes in the amount and distribution of fluorophores and in the chemico-physical properties of their environment [19]. In particular, modifications of collagen, elastin and NADH seem to account for the alteration of fluorescence observed in malignant tumors. Therefore, evaluation and measurement of AF could be a promising technique for non-invasive diagnosis and management of NMSCs [20].

Gross architectural alterations (e.g., hyperkeratosis, neoangiogenesis) associated with hyper- and hypofluorescence in pathologic tissues from the oral mucosa have been recently described [21]. Only a few other papers have been published in this field, and, to our knowledge, this is the first work on NMSC autofluorescence and its histopathological correlation [22]

The aims of the present study are 1. to investigate the differences in AF intensity between NMSC and healthy skin; 2. to analyze the histopathological determinants possibly accounting for AF alterations in pathologic skin. 

## 2. Materials and Methods

### 2.1. Study Setting and Design

The study was conducted at the Policlinico of Modena, University of Modena and Reggio Emilia (Modena, Italy) from October 2019 to March 2020.

The study was a prospective, mono-centric, pre-clinical ex vivo study approved by the local Ethical Committee (n°1167.2018 Area Vasta Emilia Nord Ethical Committee, Policlinico di Modena, Via Largo del Pozzo 71, 41124). The research was conducted according to the Helsinki Declaration. All patients signed informed consent to the use of their data for research purposes.

### 2.2. AF Detection System 

The AF detection system, (Figure 1) included:

(a)A LED source (LLS, Ocean Optics^®^, Dunedin, FL, USA), emitting light in the violet spectrum (400–430 nm), used to elicit AF in healthy and pathological skin;(b)Dichroic filters (Thorlabs^®^, Newton, NJ, USA), used to select wavelengths either from the LED source and from captured skin AF;(c)aA bifurcated optical fiber (Ocean Optics^®^) to illuminate tissues and capture the AF;(d)A spectrophotometer (Flame, Ocean Optics^®^), used to measure the spectrum of the captured AF emission light in the range;(e)A software application (SW) was also developed (in C language) and integrated into the AF detection system. The SW acquired and processed the captured AF spectra, which were then organized within a database (Figure 1).

### 2.3. Study Population and AF Evaluation

Seventy-five (46 males (61%); 29 females (39%); mean age: 79 years (ranging from 42 to 93) consecutive subjects underwent preemptive routine skin surgery for lesions suggestive for NMSC. For every patient, clinical history was considered, in particular, smoking, diabetes, hypertension, autoimmune diseases and anticoagulant therapy.

Skin phototype was as follows: type II in 71 cases (95%) and type III in 4 cases (5%). Ten (13%) patients were smokers. Overall, 98 lesions were excised from the population study. 

Forty-three (43.9%) lesions were located in the H-zone (high-risk zone: eyelids, eyebrow, periorbital region, nose, lip, chin, mandibular region, preauricular and retroauricular region, ear, temple, genitalia, hands and feet), 30 (30.6%) in the I-zone (intermediate-riskzone: cheeks, forehead, scalp, neck and pretibial region) *(n* = 30) and 25 (25.5%) in L-zone (low-risk zone: trunk and extremities).

One-hundred and fifteen lesional points were identified on the 98 excised specimens (ex vivo), and for each of these, a control point on clinically perilesional healthy skin was evaluated.

Lesional and control points were irradiated with a 400–430 nm excitation wavelength, and the emitted light was filtered in order to remove the reflected excitation light and to selectively isolate the AF emission. AF emission spectrum was measured by a spectrophotometer and then recorded. 

The ratio between the number of photons emitted at 500 nm in the healthy skin and that emitted in the pathological tissue was utilized as a measure of AF (autofluorescence intensity ratio AFIR). 

AFIR > 1 indicates a decrease in AF, while AFIR < 1 indicates an increase in AF.

After evaluation, all points were marked through suture stiches as reference for histologic analysis (Figure 2). 

Histologic analysis of the marked points evaluated was performed and on the basis of histopathological diagnosis. BCCs were subclassified into: superficial, nodular, morpheiform and pigmented. Lesions in the SCC group were subclassified into in situ SCC, Bowen disease, infiltrative SCC and keratoacanthoma. Actinic keratoses were included in the SCC group.

### 2.4. Histological Variables

The histopathological determinants evaluated for each diagnosis of NMSC and their degrees were the following: Hyperkeratosis (absent (0), mild (1), moderate (2) and severe (3));epithelial thickening (defined as an increased distance between the corneum and the basal layer) (absent (0), mild (1), moderate (2), severe (3) and with presence of ulcer (4));Fibrosis (absent (0) and present (1));Elastosis (absent (0) and present (1));Neovascularization (absent (0), mild (1), moderate (2) and severe (3));Cellular atypia (absent (0), present with grade 1, 2 or 3).

Each histopathological variable was correlated to AFIR. 

### 2.5. Statistical Analysis

Data analysis was performed using the commercial package IBM SPSS Statistics for Windows (version 22, IBM Corp., Armonk, NY) and the open-source statistical system Jamovi v.1.8.4 (The jamovi project (2021). jamovi. (Version 1.8) [Computer Software]. Retrieved from https://www.jamovi.org, accessed on 30 June 2021, based on the R system. Measures of central tendency, dispersion and shape were calculated for all the variables in the data set. Summaries included arithmetic mean, median, standard deviation, interquartile range, minimum, maximum, asymmetry, kurtosis and the relevant standard errors and 95% confidence intervals. Normality of the data was tested by the Shapiro–Wilk test. Categorical data were reported in frequency tables and expressed as absolute, relative and cumulated frequencies and percentages.

Univariate comparisons between continuous variables were performed using both parametric tests (Student’s *t*-test, ANOVA) and non-parametric test (Mann–Whitney’s *U*-test, Kruskal–Wallis test). Comparisons between categorical variables in contingency tables were performed using the chi-square test and Fisher’s exact test. 

The combined effects of different predictors on outcome variables were tested by multiple linear regression. The results were considered statistically significant for a *p*-value less than 5% (*p* < 0.05).

## 3. Results

### 3.1. AF Measurement in NMSC and Healthy Skin 

The number of excised lesions suggestive of NMSC was 98. 

All the control points on healthy skin showed an emission spectrum ranging between 450 and 700 nm (peak at 500 nm). Evaluation of all the 115 lesional points highlighted an alteration in AF (mean: 4.433; median: 3.3277). Histological analysis confirmed that almost all the areas characterized by the loss of AF were NMSC, subclassified as follows: 59 (51.3%) BCC, 53 (46.1%) SCC, and 3 (2.6%) benign lesions. Each point clinically indicated as control received a histological description of “healthy skin” (Figure 3A,B). 

All the BCC cases (100%) and 47 (90.4%) SCC cases out of 53 showed a decrease in AF. 

In particular, the mean AFIR of BCC was 4.5, indicating that, at 500 nm, BCCs are in average 4.5 times less fluorescent than healthy skin. Similarly, mean AFIR of SCC was 4.4. (Figure 4). 

No statistically significant differences between the mean of AFIR in BCC and SCC groups were detected (Student’s *t p* = 0.170; Mann–Whitney U *p* = 0.143). 

AFIR of BCC and SCC histological subtypes as well as the lowest and highest values recorded are summarized in Table 1 and Table 2.

### 3.2. Histopathological Results

#### 3.2.1. Fibrosis and Elastosis 

Fibrosis was statistically associated with alteration of AFIR. The mean AFIR varied from 3.3 to 4.7 in cases of fibrosis presence (*p* = 0.047) (Table 3 and Table 4).

A similar variation in AFIR was observed in the presence of elastosis, even if no statistical association could be highlighted (see Table 5 and Table 6). 

In cases where fibrosis and elastosis coexisted, AFIR increased, reaching a value of 7.7.

#### 3.2.2. Hyperkeratosis

In the case of absence of hyperkeratosis, mean AFIR was 4.8; when hyperkeratosis was scored as 1, 2 and 3, mean AF intensities were 4.4, 4.5 and 1.5, respectively. The Kruskal-Wallis test showed a statistically significant difference among the groups (*p* = 0.034) (Table 7). Pairwise comparisons revealed that the significance was imputable to the absence and grade 3 hyperkeratosis (*p* = 0.016) (Table 8).

#### 3.2.3. Epithelial Thickening

In cases without epithelial thickening, the AFIR was 4.2; in grade 1 thickening, 3.9; in grade 2, 4; in grade 3, 2.4; and in grade 4, 6.5. The Kruskal-Wallis test showed a statistically significant difference among the groups (*p* = 0.038) (Table 9). In particular, pairwise comparisons revealed that significant differences might be between absence and grade 4 epithelial thickening (*p* = 0.056) (Table 10).

#### 3.2.4. Neovascularization

In cases without neovascularization the mean AFIR was 2.1; in grade 1, 4.6; in grade 2, 6.4; and in grade 3, 4.8. The Kruskal-Wallis test showed a statistically significant difference among the groups (*p* = 0.019) (Table 11). In particular, pairwise comparisons revealed statistically significant differences between absent neovascularization and grade 1 neovascularization (*p* = 0.016) (Table 12).

#### 3.2.5. Cellular Atypia

In cases without cellular atypia, AFIR was 4.5; in grade 1 atypia, 3.8; in grade 2, 5.8; and in grade 3, 4.4. Kruskal-Wallis test was not statistically significant (*p* = 0.334) (Table 13). 

### 3.3. Multivariate Analysis

Multivariate analysis by a multiple linear regression model was used to analyze all histological determinants together in order to evaluate which were significant in influencing AFIR (Table 14).

When considered together, significant variables for AFIR were (presence/absence of) fibrosis, elastosis, neovascularization (absent versus grade 1, 2 and 3) and hyperkeratosis (absent versus grade 3). Epithelial thickening and cellular atypia were not statistically significant according to the linear regression model.

Clinical variables were also included in the regression, but none of these (smoking, diabetes, hypertension, autoimmune diseases or anticoagulant therapy) were significantly correlated with the degree of fluorescence of the lesion. A statistically significant correlation emerged between hypertension and elastosis (*p* = 0.021), since 85% of patients had both hypertension and elastosis 

## 4. Discussion

Despite the development of several adjunctive diagnostic aids which can possibly help the clinician in the pre-operative differentiation between benign and malignant lesions, the definitive diagnosis of the vast majority of skin conditions requires histopathological analysis. In case of doubts, the collection of one or more specimens through biopsies is the gold-standard procedure that allows the planning of an appropriate treatment [23].

As some malignant tumors (e.g., NMSCs in the early phases of development) can mimic benign lesions, the availability of a reliable, non-invasive, highly specific and sensible tool able to identify peculiar features suggestive of malignancy would provide strong input to the management of skin diseases.

Within this context, and following previous analyses performed on other anatomical sites (e.g., oral cavity) [21,24,25], we designed and conducted the present study, which attempts to evaluate the usefulness of AF for the diagnosis and the management of a heterogeneous group of NMSCs.

To date, different excitation wavelengths have been applied to fluorescence spectroscopy of the skin and in particular high-energy wavelengths in the UV region. It is known that wavelengths in the UV region, in addition to AF generation, induce oxidative stress in cells, resulting in modifications in the cellular redox state, damage to the cell structures and cell death [19]. Based on recent observation, for the present study, we selected a 400–430 excitation wavelength in the blue region of visible light [26,27].

The results of the present analysis demonstrate that all ex vivo specimens harboring neoplastic alterations at the histological level show a marked alteration of AF compared to perilesional, histologically healthy skin.

Such alterations can take the form of AF enhancement or AF reduction and are invariably present, conferring a specific optical appearance of NMSCs. In the present analysis, the vast majority of cases displayed a marked decrease in AF, which was on average 4.4–4.5 times lower than that of clinically healthy skin.

The decrease in AF in NMSCs after 400–430 excitation are in line with data already present in the literature [13,28,29,30]. In 2002, Panjehour et al. highlighted that healthy skin exhibited a stronger fluorescence emission than BCC and SCC when excited at 410 nm [31]. 

Assessment of AFIR in different BCCs and SCCs histological subtypes showed a trend to decline not statistically significantly but directly proportional to tumor thickness and infiltration. This suggests a different fluorophore distribution between healthy and lesional skin, in line with previous findings [20,29,30,32].

The histological analysis showed a correlation between histological variables and AF alteration. Our data show that fibrosis and elastosis (present vs. absent), neoangiogenesis (absent vs. present with grades 1, 2 and 3), hyperkeratosis (absent versus present with grade 3) and epithelial thickening (absent vs. present with grades 1, 2, 3 and 4) were significantly associated with alteration of AF. Cellular atypia was not significantly associated with alteration of AF.

As previously demonstrated by clinical and histological studies conducted on the oral mucosa the main fluorophores within the range of 400–460 nm are NADH and FAD cellular coenzymes, collagen and elastin [33,34]. These data may support the hypothesis that the positive association between the presence of fibrosis and elastosis with the decrease in AF can be due to alteration in the distribution and concentration of these fluorophores [32,33].

The presence of epithelial thickening, conditions characterized by an increased distance between the corneum and the basal layer, is associated with a higher decrease in AF when compared to the normal epidermis. A thickened epithelium can probably obstruct the penetration of specific wavelengths in deep dermal layers, where the main known fluorophores are located. 

Last, the presence of neovascularization was also associated with a decrease in AF. In this case, the decrease of AF can be influenced by hemoglobin reabsorption of the fluorescence from deeper dermal layer [35].

Previous studies demonstrated the correlation between AF intensity and skin phototype. Fluorescence-based methods have higher specificity and sensibility when applied to fair-skinned people [31]. Our study population is rather homogeneous with a phototype II in almost all cases evaluated (95%), thus allowing an unbiased interpretation of the results.

As extensively described in the literature, other “optical biopsy” techniques such as dermoscopy and confocal microscopy have been demonstrated to improve diagnostic accuracy in NMSC [17,36], However, dermoscopy and/or confocal imaging are used for the assessment of surgical margins either for BCC and lentigo maligna, while no evidence still exists about potential benefits of these techniques for SCCs [15,16]. 

## 5. Conclusions

Our results suggest that the analysis of fluorescence spectra could be a reliable tool for the non-invasive identification of NMSCs. Such tools might also find application for the definition of surgical margins.

To the best of our knowledge, this is the first study that systematically analyzes the variation of AFIR in relation to a series of histological variables.

## Figures and Tables

**Figure 1 cancers-13-03974-f001:**
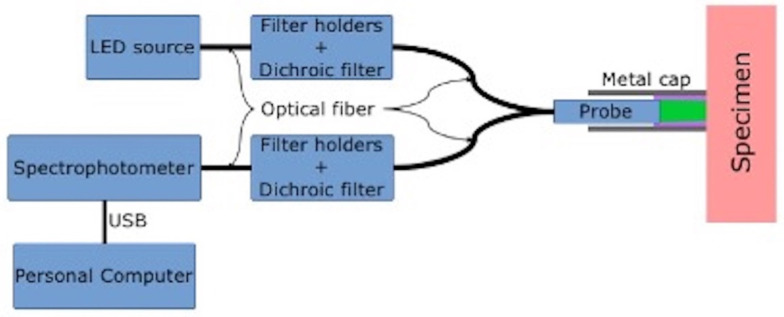
Basic representation of the AF detection system.

**Figure 2 cancers-13-03974-f002:**
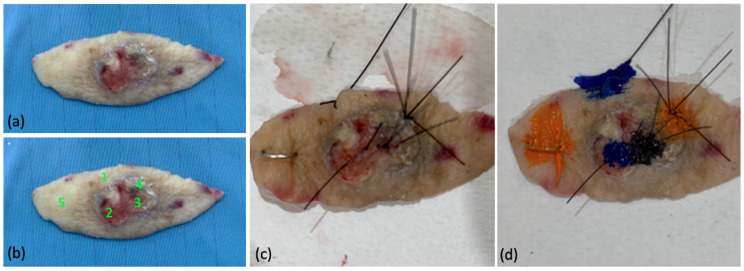
Example of lesional and healthy skin reference points on excised specimen: (**a**) specimen after excisional biopsy; (**b**) ex vivo identification of control points (1,5) and lesional points (2,3,4) for 400–430 nm irradiation; (**c**) suture stitches as reference for histologic analysis, (**d**) tissue marking dyes before histopathological examination.

**Figure 3 cancers-13-03974-f003:**
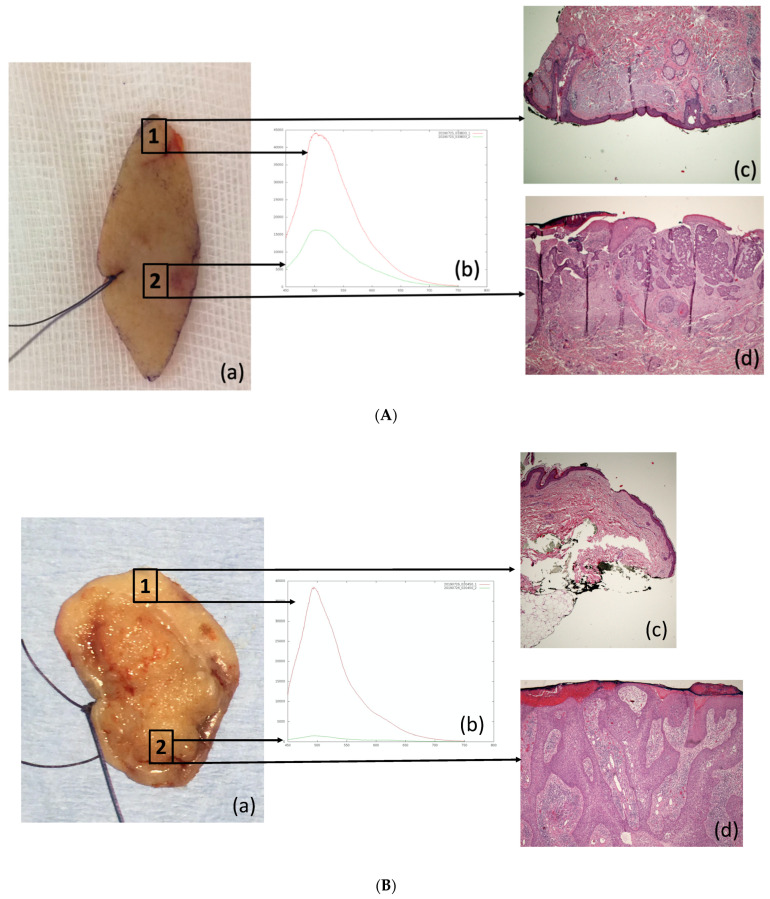
(**A**): Example of clinicopathological correlation of AF alteration in BCC: (a) identification of control point (1) and lesional point (2) on the excised specimen (b) AF emission spectra of point 1 (red line) and point 2 (green line) (c) histopathological section (hematoxylin and eosin, 4× magnification) corresponding to point 1 healthy skin, (d) histopathological section (hematoxylin and eosin, 4× magnification) corresponding to point 2 (infiltrating BCC). (**B**): Example of clinicopathological correlation of AF alteration in SCC: (a) identification of control (1) and lesional points (2); (b) AF emission spectra of point 1 (red line) and point 2 (green line); (c) histopathological section (hematoxylin and eosin, 4× magnification) corresponding to point 1 (healthy skin), (d) histopathological section (hematoxylin and eosin, 4× magnification) corresponding to point 2 (poorly differentiated SCC).

**Figure 4 cancers-13-03974-f004:**
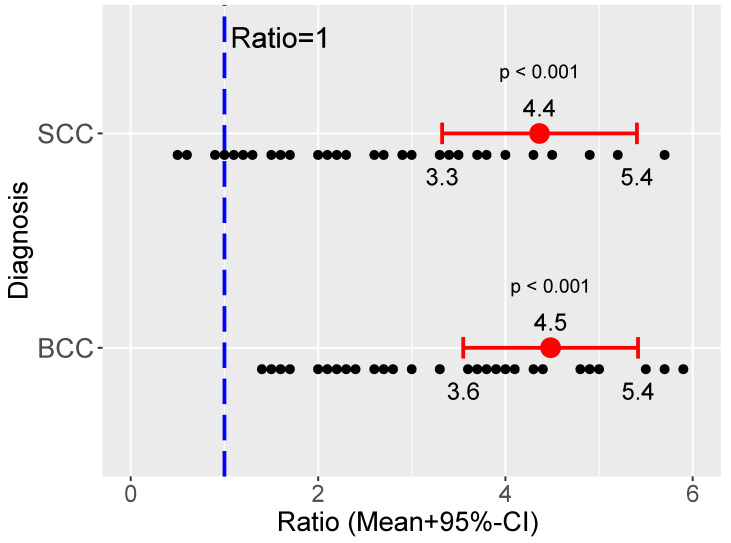
Autofluorescence Intensity Ratio in a group of NMSC (BCC and SCC).

**Table 1 cancers-13-03974-t001:** Mean AFIR among different BCC subtypes.

BCC Subtype	Mean AFIR Ratio	Minimum	Maximum	Standard Deviation	Standard Error
Superficial	2.6	1.435	5.749	1.358	0.4802
Nodular	4.4	1.354	16.10	3.437	0.5014
Morpheiform	4.8	3.607	6.082	1.750	1.237
Pigmented	12.8	9.908	15.73	4.120	2.913

**Table 2 cancers-13-03974-t002:** Mean AFIR among different SCC subtypes.

SCC Subtype	Mean AFIR Ratio	Minimum	Maximum	Standard Deviation	Standard Error
Keratoacanthoma	1.8	1.296	2.240	0.6674	0.4719
In situ SCCs	2.3	1.250	3.286	0.7685	0.2717
Bowen disease	5.0	1.342	12.98	4.173	1.391
Infiltrative SCCs	5.1	0.5689	15.63	3.482	0.6723
Actinic keratoses	4.1	0.4823	18.02	5.805	2.052

**Table 3 cancers-13-03974-t003:** Descriptive statistics for the variable “Fibrosis”. AFIR: autofluorescence intensity ratio; SD: standard deviation; SE: standard error.

Fibrosis	N	Mean AFIR	Median AFIR	SD	SE
Absent	16	3.309	2.164	3.13	0.7825
Present	99	4.676	3.521	3.734	0.3753

**Table 4 cancers-13-03974-t004:** Independent samples comparisons (Absent, Present) for the variable “Fibrosis”.

Test	Statistic	df	*p*
Student’s *t*	−1.386	113	0.168
Mann–Whitney *U*	546		0.047

**Table 5 cancers-13-03974-t005:** Descriptive statistics for the variable “Elastosis”. AFIR: autofluorescence intensity ratio; SD: standard deviation; SE: standard error.

Elastosis	*N*	Mean AFIR	Median AFIR	SD	SE
Absent	89	4.23	3.295	3.322	0.3521
Present	26	5.361	3.054	4.659	0.9138

**Table 6 cancers-13-03974-t006:** Independent samples comparisons (Absent, Present) for the variable “Elastosis”.

Test	Statistic	df	*p*
Student’s *t*	−1.386 ᵃ	113	0.169
Mann–Whitney *U*	1070		0.563

ᵃ Levene’s test is significant (*p* < 0.05), suggesting a violation of the assumption of equal variances.

**Table 7 cancers-13-03974-t007:** Kruskal-Wallis test and pairwise comparisons for the variable “hyperkeratosis”.

Kruskal-Wallis Test—Hyperkeratosis		
χ²	df	*p*
8.686	3	0.034

**Table 8 cancers-13-03974-t008:** Dwass–Steel–Critchlow–Fligner pairwise comparisons (intensity ratio) for the variable “hyperkeratosis” after the statistically significant kruskal-Wallis test. Column 1 and Column 2 represent the paired hyperkeratosis scores in comparison.

Hyperkeratosis (Score)	Hyperkeratosis (Score)	W	*p*
0	1	−0.271	0.998
0	2	−0.011	1.000
0	3	−4.189	0.016
1	2	0.0484	1.000
1	3	−3.221	0.103
2	3	−3.631	0.050

**Table 9 cancers-13-03974-t009:** Kruskal-Wallis test for the variable “epithelial thickening”.

Kruskal-Wallis Test—Epithelial Thickening		
χ²	df	*p*
10.16	4	0.038

**Table 10 cancers-13-03974-t010:** Dwass–Steel–Critchlow–Fligner pairwise comparisons (intensity ratio) for the variable “epithelial thickening” after the statistically significant kruskal-Wallis test. Column 1 and Column 2 represent the paired epithelial thickening grades in comparison.

Epithelial Thickening (Grade)	Epithelial Thickening (Grade)	W	*p*
0	1	−0.994	0.956
0	2	−0.040	1.000
0	3	−0.964	0.961
0	4	3.795	0.056
1	2	0.701	0.988
1	3	−1.000	0.955
1	4	3.200	0.157
2	3	−0.706	0.988
2	4	3.638	0.076
3	4	2.423	0.426

**Table 11 cancers-13-03974-t011:** Kruskal-Wallis test for the variable “neovascularization”.

Kruskal–Wallis Test—Neovascularization		
χ²	df	*p*
9.989	3	0.019

**Table 12 cancers-13-03974-t012:** Dwass–Steel–Critchlow–Fligner pairwise comparisons (intensity ratio) for the variable “neovascularization” after the Table 1, and Column 2 represent the paired neovascularization grades in comparison.

Neovascularization (Grade)	Neovascularization (Grade)	W	*p*
0	1	4.200	0.016
0	2	3.245	0.099
0	3	2.752	0.209
1	2	1.646	0.65
1	3	−0.751	0.952
2	3	−1.380	0.763

**Table 13 cancers-13-03974-t013:** Kruskal-Wallis test for “cellular atypia” variable.

Kruskal–Wallis Test—Cellular Atypia		
χ²	df	*p*
3.396	3	0.334

**Table 14 cancers-13-03974-t014:** Results of multiple linear regression including all histological variables as predictors. Dependent variable: AF intensity ratio.

Predictor	Estimate	SE	t	*p*
Intercept	−1.36559	1.633	−0.83614	0.405
(1) Fibrosis:				
present–absent	2.95274	1.331	2.21771	0.029
(2) Elastosis:				
present–absent	2.67802	1.071	2.50069	0.014
(3) Neovascularization:				
1–0	2.69778	1.098	2.45708	0.016
2–0	4.17302	1.539	2.71197	0.008
3–0	3.00542	1.339	2.24429	0.027
(4) Epithelial thickening:				
1–0	−0.28104	1.332	−0.21107	0.833
2–0	−0.50670	1.298	−0.39037	0.697
3–0	−1.04651	2.774	−0.37731	0.707
4–0	0.97872	1.033	0.94719	0.346
(5) Hyperkeratosis:				
1–0	−0.04351	1.260	−0.03454	0.973
2–0	−0.63988	1.015	−0.63038	0.530
3–0	−3.42011	1.714	−1.99522	0.049
(6) Cellular atypia:				
1–0	1.06353	1.178	0.90300	0.369
2–0	1.46131	1.174	1.24451	0.216
3–0	0.71902	1.455	0.49424	0.622

## Data Availability

Data available on request from the authors.

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
