# Peer review of "Correlation between Autofluorescence Intensity and Histopathological Features in Non-Melanoma Skin Cancer: An Ex Vivo Study"

_cancers, 2021, doi:10.3390/cancers13163974_

Round 1

Reviewer 1 Report

In this manuscript authors data suggest that tissue alterations could be responsible for the difference in AF emission between neoplastic and healthy tissue and these results support the potential application of AF as a useful non-invasive diagnostic tool for NMSCs.   This is well written, manuscript with appropriate methodologies to provide a conclusive results.  There are some minor concerns/suggestions that can significantly improve the manuscript.

The correlation between AFIR and histopathological features are very convincing.  Is it possible to analyze whether alterations in AFIR correlates with other well-established optical properties of skin, such as dermoscopy, reflectance confocal microscopy (RCM), multiphoton microscopy, fluorescence evaluation, diffuse reflectance and Raman spectroscopy?

Author Response

Thank you for your kind suggestion. To our knowledge, this is the first study so far on the correlation between AF and histopathological features in the skin. Dermoscopy and RCM are extensively used for the diagnosis of pigmented lesions while AF could be very useful for the diagnosis of Non Melanoma Skin Cancer and to better delineate preoperatively the surgical margins of Squamous Cell Carcinoma. We hope to have the chance in the future to match AF properties and dermoscopy or RCM specific patterns in order to improve the diagnosis and therapy of NMSC.

Reviewer 2 Report

The authors present a paper about "Correlation between autofluorescence intensity and histopathological features in non-melanoma skin cancer: an ex vivo study".

The concept of the study is innovative and the methodology followed is correct.

I have no major ehtical concerns.

The results are clearly presented and the discussion is adequate witnessing that the authors have a proper knowledge of the topic.

The conclusions are balanced.

I suggest to add some reference about the alteranative role of interventional radiotherapy whn the auhtos talk about the possible theraputic options since they only mention surgery and the most recent guidelines have been updated regarding alternative treatment options to surgery especially in the H zone (some relevant and recent data may be found in "Non-melanoma Skin Cancer Treated by Contact High-dose-rate Radiotherapy (Brachytherapy): A Mono-institutional Series and Literature Review" doi: 10.21873/invivo.12505).

I suggest to correct a few typos before publication.

Author Response

Thank you for your comment. We added the suggested reference to the text and we modified the text according to the reviewer’s kind comments.